# Effects of Dietary Inclusion of a Crude Protein Source Exhibiting the Strongest Attractiveness to Red Sea Bream (*Pagrus major*) on Growth, Feed Availability, and Economic Efficiency

**DOI:** 10.3390/ani14050771

**Published:** 2024-02-29

**Authors:** Seong Il Baek, Sung Hwoan Cho

**Affiliations:** 1Department of Convergence Study on the Ocean Science and Technology, Korea Maritime and Ocean University, Busan 49112, Republic of Korea; qortjddlff@naver.com; 2Division of Convergence on Marine Science, Korea Maritime and Ocean University, Busan 49112, Republic of Korea

**Keywords:** attractiveness, jack mackerel meal, growth performance, feed availability, economic profit index

## Abstract

**Simple Summary:**

Crude feed protein sources, which are rich in several amino acids, can be used as feed attractants when formulating feed. Jack mackerel meal exhibited the strongest attractiveness to red sea bream among 18 crude protein ingredients. The effects of the inclusion of various levels of jack mackerel meal in diets on the growth and feed availability of red sea bream were evaluated in an 8-week feeding trial. The results of the 8-week feeding trial proved that the dietary inclusion level of 100% jack mackerel meal in a 60%-FM-based diet produced the greatest growth performance in red sea bream, which directly resulted from the improved feed consumption. This result may imply the potential use of jack mackerel meal as a feed attractant when developing low-fish-meal diets for the sustainable culture of red sea bream.

**Abstract:**

Dietary incorporation of an attractive feed protein source is a practical method of enhancing feed intake and consequently improving the growth of fish. The attractiveness of 18 crude protein sources to juvenile red sea bream (*Pagrus major*) and the effects of the dietary inclusion of the crude protein source that exhibited the strongest attractiveness on growth, feed availability, and economic efficiency were determined. Jack mackerel meal (JMM) showed the strongest attractiveness to red sea bream among 18 crude protein ingredients. In an 8-week feeding trial, 810 juveniles were randomly distributed into 27 tanks (30 fish/tank). Nine experimental diets were prepared. The control (Con) diet included 60% fish meal (FM). Various levels (1, 3, 5, 10, 20, 40, 60, and 100%) of JMM were included at the expense of FM in the Con diet, and the resulting diets were named the JMM_1_, JMM_3_, JMM_5_, JMM_10_, JMM_20_, JMM_40_, JMM_60_, and JMM_100_ diets, respectively. Fish were hand-fed to apparent satiation twice daily for 8 weeks. The weight gain, specific growth rate, and feed consumption of red sea bream that were fed the JMM_40_, JMM_60_, and JMM_100_ diets were significantly (*p* < 0.0001 for all) higher than those of the fish fed all other diets. However, dietary JMM inclusion had no remarkable impacts on the feed utilization, biological indices, and chemical composition of the whole body of red sea bream. In terms of the economic view of the study, the economic profit index of red sea bream fed the JMM_40_, JMM_60_, and JMM_100_ diets was significantly (*p* < 0.0001) greater than that of the fish fed all other diets. In conclusion, the strongest attractiveness to red sea bream among 18 crude protein sources was observed in JMM. The inclusion of more than 40% JMM at the expense of FM in the diet of red sea bream is highly recommended for practical feed formulations to induce remarkable improvement in the growth performance of fish and the economic returns for farmers.

## 1. Introduction

In aquaculture, feeding is the primary means through which fish obtain nutrition and energy, providing a material and energy foundation for their various life-sustaining functions, including their survival, growth, and reproduction [1,2]. The highest fish growth rate, which is generally associated with the highest feed consumption and feed efficiency in aquaculture operations, indeed leads to maximum profitability for fish farmers [3,4]. Since feed is one of the costliest input factors and accounts for over 50% of the total production cost in aquaculture operations, improving feed consumption and minimizing feed wastage are very crucial for successful and sustainable aquaculture [5]. 

To elicit the strongest feeding response in a target fish species, the formulated diet must have the appropriate size, shape, color, texture, and density (buoyancy), as well as high attractiveness and palatability to the fish [6]. To improve feed consumption by fish, a variety of feed attractants have been commonly used in aquafeeds. Feed attractants are commonly low-molecular-weight metabolites, such as amino acids (AAs), quaternary ammonium compounds, nucleosides, nucleotides, and organic acids, which are vital tissue constituents of major (common) prey and make up the natural feed [7,8]. Over the last few decades, considerable attention has been paid to the inclusion of feed attractants and/or stimulants in formulated aquafeeds [9]. Their application has been found to improve feed palatability, enhance feed consumption, accelerate growth performance, minimize feed wastage and water pollution in surroundings, and, ultimately, lower production costs [6,10,11].

The fish-feeding processes are controlled by olfactory and gustatory systems, which assist in finding an original location (olfactory system) and determining the final feed consumption (gustatory system) [12]. Some AAs are known as a significant class of olfactory cues that cause a variety of fish to engage in feeding behavior [13,14]. Some carnivorous fish species exhibit positive responses to alkaline and neutral nitrogenous compounds, such as glycine, proline, taurine, valine, and betaine [15]. Carr et al. [16] revealed that tissue extracts from 10 marine fish species and 20 species of invertebrates contained high levels of five commonly cited feed attractants (glycine, alanine, proline, arginine, and betaine) and highlighted that glycine and alanine were two major tissue constituents based on the feeding behavior of 35 fish species. AA fractions were also found to exhibit attraction responses in the nerves involved in the olfactory and gustatory systems of Atlantic halibut (*Hippoglossus hippoglossus*) [14]. Further, several studies proved that dietary supplementation with attractants, such as specific AAs, betaine, and nucleotides, improved feed consumption by fish [11,17,18,19].

However, the classification of whether these chemical substances used as attractants in diets truly attract target fish is still controversial in practical feeding, and the acceptability and palatability of diets supplemented with these chemicals for fish cannot be easily explained by individual attractants [8,20]. Instead, each attractant showed stronger feeding stimulation when supplemented as a mixture compared to that when being supplemented individually [8]. For instance, Kohbara et al. [21] reported that the inclusion of a synthetic chemical mixture (AAs, nucleotides, and nucleosides) was inferior to the use of natural feed stimulants (extracts of jack mackerel muscle) for yellowtail (*Seriola quinqueradiata*) based on the numbers of pellets swallowed by fish. In accordance with this, tissue extracts of aquatic animals, such as fish [22], shrimp [23], squid [24,25,26], krill [24], mussels [27,28], and marine worms [29], have been used as effective natural feed stimulants. In addition, several studies have also proved that crude protein sources that are commonly used in aquafeeds can be used as attractants and/or stimulants, and their dietary inclusion consequently enhances feed consumption [9,30,31,32,33,34,35]. Therefore, the attractiveness of various crude protein sources to target fish species and their nutritional values need to be evaluated prior to their inclusion in diets [36].

Red sea bream (*Pagrus major*) is considered the representative aquaculture fish species in the Republic of Korea (hereafter, Korea) because of its high commercial market value and consumer demand [37,38]. The annual aquaculture production of red sea bream in Korea reached 8078 metric tons in 2022 [39]. Because of its commercial importance, numerous studies have been performed to elucidate its dietary nutritional requirements [40,41,42], the dietary replacement effects of alternative sources of fish meal (FM) [34,43,44], and the effects of the dietary inclusion of functional additives [45,46]. In addition, Kader et al. [47] unveiled that the inclusion of 10% krill meal, 10% squid meal, and 2.5% AAs with 10% soluble fish and 15% of their blend in low-FM diets replacing 60% FM with soy protein concentrate achieved comparable, and superior growth to that of juvenile red sea bream fed a 60%-FM-based diet, respectively. Later, the growth of red sea bream that were fed diets replacing 70 and 80% FM with dehulled soybean meal supplemented with a crude attractant blend (10% fish soluble, 5% krill meal, and 5% squid meal) outperformed that of fish fed a 60%-FM-based diet [34]. Therefore, the manipulation of crude feed ingredients exhibiting high attractiveness to target fish in feed can effectively improve the growth resulting from improved feed intake. 

Our earlier studies determined the attractiveness of various sources of feed ingredients to olive flounder (*Paralichthys olivaceus*) [32,33] and rockfish (*Sebastes schlegeli*) [9,48] and proved that the dietary inclusion of the protein source that was most attractive to both fish, jack mackerel meal (JMM), ameliorated their growth performance, which was directly reflected by their enhanced feed consumption. To the best of our knowledge, however, no studies have been conducted to evaluate the attractiveness of protein sources to red sea bream. The purpose of this study was, therefore, to evaluate the attractiveness of various protein sources to red sea bream and then to evaluate the effects of the dietary inclusion of the most attractive protein source on the growth, feed utilization, chemical composition, and economic efficiency of juvenile red sea bream.

## 2. Materials and Methods

### 2.1. Measurement of the Attractiveness of Various Crude Protein Sources to Red Sea Bream

#### 2.1.1. Protein Sources Used to Evaluate Attractiveness to Red Sea Bream

Eighteen crude protein sources (5 kinds of FM, 3 kinds of crustacean meals, 3 kinds of mollusk meals, 3 kinds of animal byproduct meals, and 4 kinds of plant protein meals) were tested to evaluate their attractiveness to juvenile red sea bream, and their chemical compositions and suppliers are presented in Table 1. 

#### 2.1.2. Preparation of Red Sea Bream

The juvenile red sea bream used for the determination of the attractiveness of the crude protein sources were purchased from a private hatchery (Tongyeong-si, Chungcheongnam-do, Korea) and acclimatized to a 5-ton round-shape flow-through tank. Before the test, the fish were fed with a commercial pellet (50% crude protein and 8% crude lipid) (National Federation of Fisheries Cooperatives Feed, Uiryeong-gun, Gyeongsangnam-do, Korea) twice per day for two weeks. Similar sizes of juvenile sea bream [41.0–46.9 g (43.9 g ± 1.31 g; means ± SD)] were used to determine the attractiveness of the 18 protein sources in a preliminary test (1st–8th tests), and then the top four protein sources (9th and 10th tests) were chosen to finally elucidate the strongest attractiveness of the crude protein sources to juvenile fish.

#### 2.1.3. Preparation of Equipment for Evaluating the Attractiveness of Crude Protein Sources to Red Sea Bream

Three sets of reinforced and flow-through acrylic transparent tanks (1.0 × 0.6 × 0.5 m; water volume: 300 L) made of three equally divided rectangular attracting chambers (0.6 × 0.2 × 0.5 m each) and an acclimatization chamber (0.4 × 0.6 × 0.5 m) were used to determine the attractiveness of the various crude protein sources to juvenile red sea bream (Figure 1). 

The flow rates of the three attracting chambers were kept equal (1.6 L/min/chamber) using a portable ultrasonic flow meter (TF1100-CH, Lanry Instruments Co., Ltd., Shanghai, China). The attracting and acclimatization chambers were divided by a vertically movable acrylic shutter. Funnel-shaped attracting chambers (10 and 5 cm in radius for the entrance and exit, respectively) were designed to allow free access of the red sea bream to different crude protein sources in each attracting chamber and prevent them from swimming back to the acclimatization chamber. Movement into the attracting chamber was video-recorded to count the number of red sea bream entering through the funnel-shaped entrance. The water source was a mixture of sand-filtered seawater and underground seawater (1:1). The water temperature was monitored daily; it ranged from 21.1 to 22.5 °C (21.7 ± 0.40 °C; mean ± SD) from the 1st to the 10th test. Moderate aeration was supplied in each chamber, and the photoperiod followed natural conditions.

#### 2.1.4. Determining the Attractiveness of Crude Protein Sources to Red Sea Bream 

Three different crude protein sources at a time were randomly assigned to evaluate their attractiveness to red sea bream. A total of 25 fish of similar sizes were randomly distributed in the acclimatization chamber for 24 h without feeding before the test. Then, 30 g of different crude protein sources wrapped in a 209 μm (mesh size) micromesh screen fabric (PET 1500 32/83-100W, Sefar, Heiden, Switzerland) were placed onto the surface of the water of each attracting chamber. Then, the shutter was raised to allow red sea bream free access to each protein source in the attracting chambers for 30 min. The raised shutter was returned to its original place to count the number of red sea bream that moved toward the crude protein sources in the attracting chambers. A knockout comparison was adopted to determine the attractiveness of the protein sources to red sea bream. The number of red sea bream moving towards each attracting chamber in a 10-min interval was analyzed based on video recordings obtained during the 30 min of observation, and the attractiveness at various elapsed time intervals was compared.

The eighteen crude protein sources were compared to evaluate their attractiveness to red sea bream in each test (1st to 8th test) in a preliminary test, and then their attractiveness to red sea bream was compared to determine the most attractive one (9th to 10th test). Each test was performed in triplicate by switching the locations of the protein sources. Once the red sea bream were used to determine the attractiveness of the protein sources, they were excluded from further tests. 

### 2.2. Application of the Crude Protein Source Exhibiting the Strongest Attractiveness to Red Sea Bream in Experimental Feeds in a Feeding Experiment 

#### 2.2.1. Preparation of the Fish and Rearing Conditions

The juvenile red sea bream used for the feeding trial were purchased from a private hatchery (Tongyeong-si, Chungcheongnam-do, Korea). Prior to the feeding trial, juvenile fish were acclimatized to a 5-ton round-shape flow-through tank and fed with a commercial pellet (50% crude protein and 8% crude lipid) (National Federation of Fisheries Cooperatives Feed, Uiryeong-gun, Gyeongsangnam-do, Korea) for two weeks. After the 2-week acclimatization period, 810 juvenile fish (initial weight of 3.2 g) were randomly distributed into 27 50 L flow-through tanks with the nine experimental diets in triplicate (30 fish/tank). Each tank was supplied with a blend of sand-filtered seawater and underground seawater (1:1), and proper aeration was continuously supplied to each tank. The water temperature, dissolved oxygen, salinity, and pH were monitored daily by using a digital multimeter (AZ-8603, AZ instrument, Taichung, Taiwan), and they ranged from 12.6 to 19.2 °C [15.6 ± 2.27 °C (means ± SD)], 7.2 to 8.8 mg/L [8.1 ± 0.43 mg/L (means ± SD)], 30.3 to 33.6 g/L [32.3 ± 0.73 g/L (means ± SD)], and 6.5 to 7.4 [6.8 ± 0.33 (means ± SD)], respectively. The water flow rate in each tank was 1.6 L/min, and the photoperiod followed natural conditions. The fish were carefully hand-fed to apparent satiation twice a day (08:00 and 17:00) for 8 weeks. Feed consumption was recorded daily for each tank. Daily siphon cleaning of the bottom of tanks was performed to maintain a reasonable water quality, and dead fish were removed immediately when they were observed.

#### 2.2.2. Preparation of the Experimental Feeds

Nine experimental feeds were formulated (Table 2). Sixty-percent FM and 17.5% soybean meal were included as the protein sources in the control (Con) diet. The Con diet also contained 14.4% wheat flour and 5.6% fish oil as the carbohydrate and lipid sources, respectively. In the Con diet, graded levels (1, 3, 5, 10, 20, 40, 60, and 100%) of JMM were included at the expense of FM, and these diets were named the JMM_1_, JMM_3_, JMM_5_, JMM_10_, JMM_20_, JMM_40_, JMM_60_, and JMM_100_ diets, respectively. The ingredients of the experimental feeds were well blended and made into dry pellets using a laboratory pellet extruder (Dongsung Mechanics, Busan, Korea) equipped with a 2 mm die, and then they were air-dried at 50 °C for 24 h. The experimental feeds were formulated to fulfill the dietary nutrient requirements (52% protein and 15% lipid) for red sea bream (Takeuchi et al., 1991 [41]). The feeds were kept in a freezer at –20 °C until use. 

#### 2.2.3. Evaluation of the Attractiveness of the Experimental Feeds to Red Sea Bream

The same procedures and methods used to determine the attractiveness of the various crude protein sources to red sea bream were adopted (2.1.3 and 2.1.4 in Materials and Methods). Thirty juvenile fish of similar size [4.2–5.8 g (4.9 g ± 0.43 g; means ± SD)] were used to evaluate the attractiveness of the experimental feeds in each test (1st to 4th test) in triplicate. The water temperature changed from 19.1 °C to 20.8 °C (19.8 ± 0.55 °C; mean ± SD) throughout the 4th test.

#### 2.2.4. Measurement of the Biological Indices of Red Sea Bream 

At the termination of the 8-week feeding experiment, all surviving red sea bream in each tank were starved for 24 h and then anesthetized with tricaine methanesulfonate (MS-222) at 100 ppm. All live fish from each tank were counted and collectively weighed to evaluate their survival and weight gain. Ten red sea bream from each tank were randomly chosen to calculate their biological indices, such as their condition factor (K), viscerosomatic index (VSI), and hepatosomatic index (HSI). The following growth parameters and somatic indices were calculated: specific growth rate (SGR, %/day) = (Ln final weight of fish − Ln initial weight of fish) × 100/days of feeding trial (56 days), feed efficiency (FE) = [total final weight (g) − total initial weight (g) + total weight of dead fish (g)]/total feed consumption (g), protein efficiency ratio (PER) = weight gain of fish/protein consumption of fish, protein retention (PR, %) = protein gain of fish × 100/protein consumption of fish, K (g/cm^3^) = body weight of fish (g) × 100/total length of fish (cm)^3^, VSI (%) = viscera weight of fish × 100/body weight of fish, and HSI (%) = liver weight of fish × 100/body weight of fish.

#### 2.2.5. Chemical Analysis of the Experimental Feeds and Fish

Ten red sea bream at the beginning of the feeding trial and all remaining fish (≥15) from each tank at the termination of the feeding trial were sampled for the chemical analysis of their whole bodies. The moisture, crude protein, crude lipid, and ash contents were determined according to the standard procedures [49]. The crude protein and lipid contents were assessed using the Kjeldahl method (Kjeltec 2100 Distillation Unit, Foss Tecator, Hoganas, Sweden) and an ether extraction method (Soxtec TM 2043 Fat Extraction System, Foss Tecator, Hoganas, Sweden), respectively. The moisture content was measured by oven drying the samples for 24 h at 105 °C, and a muffle furnace was used at 550 °C for 4 h to determine the ash content. 

#### 2.2.6. Amino and Fatty Acid Analysis of the Experimental Feeds

Except for methionine, cysteine, and tryptophan, all AAs in the experimental feeds were analyzed using an AA analyzer (L-8800 Auto-analyzer: Hitachi, Tokyo, Japan), followed by ion-exchange chromatography after hydrolyzing with 6 N HCl at 110 °C for 24 h. For the analysis of methionine and cysteine, samples were oxidized with performic acid for 24 h at a temperature below 5 °C to make methionine sulfone and cysteic acid. Then, they were freeze-dried twice with deionized water, hydrolyzed, and analyzed according to the standard procedure used for the other AAs. The tryptophan content was measured with high-performance liquid chromatography (S1125 HPLC pump system, Sykam GmbH, Eresing, Germany).

The fatty acids (FAs) in the experimental feeds were extracted with a mixture of chloroform and methanol (2:1 *v*/*v*) according to [50], and FA methyl esters were prepared through transesterification with 14% BF_3_-MeOH (Sigma, St. Louis, MO, USA). FA methyl esters were analyzed using gas chromatography (Trace GC, Thermo, Waltham, MA, USA) with a flame ionization detector equipped with an SPTM-2560 capillary column (100 m × 0.25 mm I.d. film thickness 0.20 µm; Supelco, Bellefonte, PA, USA).

#### 2.2.7. Economic Analysis of the Feeding Experiment

An economic evaluation was conducted with USD as the currency. The economic conversion ratio (ECR) and economic profit index (EPI) were calculated according to [51,52]: ECR (USD/kg) = feed consumption of fish (kg/fish)/[weight gain of fish (kg/fish) × diet price (USD/kg)], and EPI (USD/fish) = [final weight of fish (kg/fish) × selling price of fish (USD/kg)] − [feed consumption of fish (kg/fish) × diet price (USD/kg)]. The prices of red sea bream and feed ingredients were calculated using the exchange rate of USD = 1232 won (Korean currency). The price of juvenile red sea bream in Korea was estimated at 20.29 USD/kg based on a fish farmer’s assessment. The price of the experimental feeds was calculated by multiplying the respective contributions of each feed ingredient by the respective cost per kg and summing the values obtained for all of the ingredients in the experimental feeds. The price (USD/kg) of each ingredient was as follows: FM = USD 2.26; JMM = USD 2.43; fermented soybean meal = USD 0.70; wheat flour = USD 0.55; fish oil = USD 2.76; vitamin premix = USD 8.28; mineral premix = USD 6.66; choline = USD 1.30.

### 2.3. Statistical Analysis

The results were analyzed using SPSS version 24.0 (SPSS Inc., Chicago, IL, USA). Data were evaluated for assumptions including normality and homogeneity of variance using the Shapiro–Wilk and Levene tests, respectively, and no violations were detected (*p* > 0.05). One-way ANOVA and Tukey’s honestly significant differences (HSD) post-hoc test were adopted to compare the means of the dietary treatments. All percentage data were arcsine-transformed before statistical analysis. Additionally, a regression analysis of the JMM levels and each AA’s content in the experimental feeds versus weight gain, SGR, and feed consumption of red sea bream was performed. The differences in the attractiveness of the crude protein sources and experimental feeds to red sea bream were tested with the Chi-square contingency test and Bonferroni’s test. *p* < 0.05 was considered statistically significant. 

## 3. Results

### 3.1. Attractiveness of Various Crude Protein Sources and Experimental Diets to Red Sea Bream

The attractiveness of the various crude protein sources to red sea bream over the elapsed time is shown in Table 3. From the first to the sixth test, jack mackerel meal (JMM), anchovy meal (AM), sardine meal (SM), squid liver meal (SLM), tuna byproduct meal (TBM), and hydrolyzed fish meal (HFM) showed the highest attractiveness to red sea bream throughout the 30 min of observation after the introduction of the crude protein sources into the attracting chamber. In the seventh and the eighth tests, JMM and SM showed the highest attractiveness to the fish throughout the 30 min of observation after the introduction of the crude protein sources into the attracting chamber. In addition, SLM and TBM also showed stronger attractiveness to the fish than that of the other protein sources (AM and HFM) in each test. Finally, in the ninth and tenth tests, the strongest attractiveness to red sea bream was observed for JMM, followed by SM, SLM, and TBM in that order, throughout the 30 min observation of (*p* < 0.0001 for all).

The attractiveness of the JMM_40_, JMM_60_, and JMM_100_ diets to red sea bream was significantly (*p* < 0.0001 for all) stronger than that of all other diets in the first, second, and third tests, respectively, throughout the 30 min of observation after the introduction of the experimental diets (Table 4). Finally, in the fourth test, significantly (*p* < 0.0001 for all) stronger attractiveness to red sea bream was observed for the JMM_100_ and JMM_60_ diets compared to the JMM_40_ diet.

### 3.2. Amino and Fatty Acid Profiles of the Experimental Diets

The AA profiles of FM, JMM, and the experimental feeds are presented in Table 5. The contents of histidine, threonine, glutamic acid, glycine, and proline in the experimental feeds proportionally increased with elevated JMM inclusion levels in the experimental feeds, but they decreased for phenylalanine and valine. 

The FA profiles of FM, JMM, and the experimental feeds are given in Table 6. The total saturated FA (∑SFA), linoleic acid (C18:2n-6), arachidonic acid (C20:4n-6), and docosahexaenoic acid (DHA, C22:6n-3) contents and DHA:EPA in JMM were relatively higher than those in FM, but they were low for total monounsaturated FA (∑ MUFA) and eicosapentaenoic acid (EPA, C20:5n-3). The contents of ∑SFA, linoleic acid, arachidonic acid, and DHA, as well as DHA:EPA, of the experimental feeds proportionally increased with elevated JMM inclusion levels in the experimental feeds, but they decreased for the ∑MUFA and EPA contents.

### 3.3. Performance of Red Sea Bream in the 8-Week Feeding Experiment

#### 3.3.1. Survival and Growth Performance of the Fish

The survival of red sea bream changed from 93.3% to 97.8%, and it was not significantly (*p* > 0.8) affected by the dietary treatments (Table 7). The weight gain and SGR of red sea bream fed the JMM_40_, JMM_60_, and JMM_100_ diets were significantly (*p* < 0.0001 for both) higher than those of the fish fed all other diets (the Con, JMM_1_, JMM_3_, JMM_5_, JMM_10_, and JMM_20_ diets). The regression analysis revealed significant quadratic relationships between the dietary JMM inclusion levels at the expense of FM versus weight gain (Y = −0.000415X^2^ + 0.07034X + 9.3732, R^2^ = 0.8992, *p* < 0.0001, Y_max_ = 84.8%) and SGR (Y = −0.000056X^2^ + 0.009198X + 2.4545, R^2^ = 0.9210, *p* < 0.0001, Y_max_ = 82.1%), respectively. 

Strong quadratic and cubic relationships of histidine, threonine, and lysine contents among the essential AAs (EAAs) and of alanine, glycine, glutamic acid, and proline among the non-EAAs (NEAAs) in the experimental feeds with the weight gain, SGR, and feed consumption of red sea bream were observed (Figure 2 and Figure 3, respectively). 

#### 3.3.2. Feed Availability and Biological Indices of Red Sea Bream 

The feed consumption of red sea bream fed the JMM_60_ and JMM_100_ diets was significantly (*p* < 0.0001) higher than that of fish fed all other diets, except for the JMM_40_ diet (Table 8). The regression analysis revealed a significant quadratic relationship between the dietary inclusion level of JMM at the expense of FM and feed consumption (Y = −0.000388X^2^ + 0.071494X + 10.0247, R^2^ = 0.9635, *p* < 0.0001, Y_max_ = X value of 92.1%).

The FE of the red sea bream ranged from 1.05 to 1.09, the PER ranged from 1.74 to 1.83, and the PR ranged from 27.6% to 30.4%. These parameters were not remarkably (*p* > 0.6, *p* > 0.4, and *p* > 0.7, respectively) altered by the dietary treatments. 

The K values of the red sea bream ranged from 1.85 g/cm^3^ to 1.88 g/cm^3^, the VSI ranged from 6.82% to 7.02%, and the HSI ranged from 1.39% to 1.47%. None of these parameters were remarkably (*p* > 0.8, *p* > 0.9, and *p* > 0.9, respectively) changed by the dietary treatments. 

#### 3.3.3. Chemical Composition of the Whole Body of Red Sea Bream

The moisture content of the whole body of red sea bream changed from 72.9% to 76.2%, the crude protein content changed from 15.3% to 16.0%, the crude lipid content changed from 9.2% to 9.6%, and the ash content changed from 4.5% to 5.2% (Table 9). None of these parameters were remarkably (*p* > 0.3, *p* > 0.8, *p* > 0.9, and *p* > 0.7, respectively) altered by the dietary treatments.

#### 3.3.4. Economic Evaluation of the Experimental Diets

The price of the experimental feeds tended to increase with the dietary JMM inclusion levels (Table 10). The ECR of the JMM_100_ diet was remarkably (*p* < 0.003) higher than that of all other experimental diets, except for the JMM_60_ diet, while the EPIs of the JMM_40_, JMM_60_, and JMM_100_ diets were remarkably (*p* < 0.0001) greater than those of all other diets. The regression analysis revealed significant linear and quadratic relationships between the dietary JMM inclusion levels at the expense of FM versus ECR (Y = 0.00263X + 1.5002, R^2^ = 0.8935, *p* < 0.0001) and EPI (Y = −0.000008229X^2^ + 0.001326X + 0.2355, R^2^ = 0.8792, *p* < 0.0001), respectively.

## 4. Discussion 

The apparatus used in the present study to evaluate the attractiveness of crude protein sources and experimental feeds to red sea bream based on their feeding behavior (movement toward the attracting chambers after their introduction into the attracting chambers) seemed to be the highly reliable and practical approaches, as they were previously applied olive flounder [32] and rockfish [9,48]. Following 24 h of feed deprivation, the red sea bream evoked relatively strong feeding behavior within 10 min after the introduction of the samples in each test. The feeding behavior of fish could mainly have been triggered by olfactory stimuli, to which the water-soluble extracts of the crude proteins’ ingredients and diets were attributed. However, the final acceptability of feed for fish is influenced by all sensory stimulus systems (vision, olfactory, and gustation stimuli) in combination [8]. The four types of crude protein sources (JMM, SM, SLM, and TBM) found to have the strongest attractiveness to red sea bream among the 18 crude protein sources in this experiment were corroborated by previous studies [56,57], and it is emphasized that marine-originating ingredients, such as FM, fish solubles, fish hydrolysates, SLM, and KM, are widely known attractants in aquafeeds due to their abundancy in nucleotides and AA content. In the following tests (ninth and tenth tests), the strongest attractiveness to red sea bream was observed for JMM, followed by SM, SLM, and TBM in that order. Likewise, jack mackerel muscle extracts—particularly inosine-5ʹ-monophosphate (IMP) and lactic acid—were regarded as highly effective feed attractants for yellowtail [22], and histidine was regarded as such for olive flounder [58]. Takakuwa et al. [59] also proved that the IMP in the muscle extracts of jack mackerel has a strong feed-stimulating effect on greater amberjack (*Trachurus japonicus*). In the determination of the attractiveness of the experimental feeds to red sea bream in this experiment, the diet containing the greatest amount of JMM always exhibited the strongest attractiveness in the first–third tests (Table 4). In the fourth and final test, both the JMM_60_ and JMM_100_ diets exhibited stronger attractiveness than that of the JMM_40_ diet throughout the 30-min observation period. These results might indicate that increased dietary JMM inclusion levels could directly improve attractiveness to red sea bream. 

After the 8-week feeding trial, red sea bream fed the JMM_100_ diet achieved the greatest growth performance, and the optimal dietary inclusion levels of JMM at the expense of FM were estimated to be 84.8 and 82.1%, respectively, for inducing the maximum weight gain and SGR of the fish based on the quadratic model in the regression analysis. However, there were no remarkable differences in weight gain and SGR among red sea bream fed the JMM_40_, JMM_60_, and JMM_100_ diets in this study. These results suggest that the inclusion of 40% JMM (24% in diet) was the lowest JMM inclusion level that could enhance the growth performance of red sea bream. This inclusion level is particularly recommended when formulating low-FM diets to reduce feed costs and maximize the productivity of red sea bream farming. The practical application of JMM in low-FM diets for red sea bream will have to be evaluated in a further study.

The superior weight gain and SGR of the red sea bream fed the JMM_40_, JMM_60_, and JMM_100_ diets with respect to the fish fed all other diets (Con, JMM_1_, JMM_3_, JMM_5_, JMM_10_, and JMM_20_ diets) were directly reflected by the higher feed consumption by the former, eventually leading to there being no difference in feed utilization (FE, PER, and PR) in the present experiment. Likewise, Jeong et al. [33] demonstrated that improved growth of olive flounder was directly reflected by increased feed consumption with increased dietary levels of JMM, eventually producing no difference in feed utilization. Dietary supplementation with tuna hydrolysate, squid liver paste, and multi-stimulant blends (*Bacillus amyloliquefaciens*, sodium butyrate, zinc methionine, and digestive enzymes) as feed stimulants in barramundi (*Lates calcarife*), genetically improved farmed tilapia (GIFT) (*Oreochromis* sp.), and common carp (*Cyprinus carpio*), respectively, also improved their growth performance, which was directly reflected in their increased feed consumption, eventually resulting in there being no difference in FE [60,61,62].

The gustatory responses of fish to AAs have been proven to vary depending on the fish species, unlike their olfactory responses [63]. Strong quadratic and cubic relationships of histidine, threonine, and lysine contents among the EAAs, and of alanine, glycine, glutamic acid, and proline among the NEAAs in the experimental diets with the weight gain, SGR, and feed consumption of red sea bream were observed in this study. These results indicated that these AAs in the experimental diets were closely associated with growth performance and feed consumption, and they acted as feed attractants for red sea bream. Likewise, strong gustatory responses were observed in red sea bream with synthetic AAs, such as alanine, glycine, arginine, and serine, and they were particularly prominent in alanine and glycine based on electro-neurophysiological studies [63]. Shimizu et al. [64] also emphasized that glycine, glucosamine, and proline in non-muscle krill meal (without abdominal muscle) provoked intense feeding stimulation in red sea bream. Similarly, strong quadratic relationships of the alanine, glycine, and histidine contents in various protein sources (shrimp meal, squid meal, pollock meal, SM, and JMM) with the weight gain and feed consumption of rockfish were found when juvenile rockfish were fed with a diet containing 5% each of shrimp meal, squid meal, pollock meal, SM, and JMM, which exhibited strong attractiveness at the expense of anchovy meal, which exhibited only moderate attractiveness to rockfish for 8 weeks [9]. In addition, the application of the combination of 0.5% glutamic acid and FM hydrolysate at 1.8 mL/kg was able to enhance the palatability of an FM-free diet containing 63% soy protein concentrate, and it led to higher feed consumption in red sea bream than that in fish fed a 60%-FM-based diet [64]. Similarly, histidine and glutamic acid showed strong feeding stimulation activity in olive flounder and Pacific bluefin tuna (*Thunnus orientalis*), respectively [58,65,66]. Considering the results of this study and previous studies, we can conclude that the increased histidine, alanine, glycine, glutamic acid, and proline contents in the experimental diets with increasing levels of JMM directly improved the feed consumption of red sea bream.

The dietary EPA and DHA requirements for the growth of juvenile red sea bream were estimated to be 1% (6.62% of total FAs) and 0.5% (3.31% of total FAs), respectively, in the diets when either the DHA or the EPA was completely absent [40]. All experimental diets met the dietary requirements for DHA, but they did not meet the dietary EPA requirements. However, Takeuchi et al. [40] also reported that their requirements could be reduced by 0.25% (1.66% of total FAs) in each diet when both were sufficiently supplied. Therefore, the somewhat insufficient content of EPA in the experimental diets seemed to have an insignificant effect on the growth of the fish in this study. Moreover, the ratio of DHA to EPA increased from 0.92 to 1.60 with the increased dietary JMM levels in this study. Past studies proved that the dietary ratio of DHA to EPA could largely influence the growth of marine fish species, such as golden pompano (*Trachinotus ovatus*) [67], sea bass (*Lateolabrax japonicas*) [68], and starry flounder (*Platichthys stellatus*) [69], and they reported that suitable dietary ratios of DHA to EPA were estimated at 1.40, 2.05, and 1.24, respectively. The proper ratio of DHA to EPA in a diet is species-specific [70], and that for red sea bream has not yet been studied. However, considering that the efficiency of DHA is twice as high as that of EPA [40], the increasing trend of the ratio of DHA to EPA in the experimental feeds with the increased dietary JMM inclusion levels seemed to bring about a positive impact on the growth performance of red sea bream. 

The biological indices of fish, such as K, VSI, and his, are generally considered critical parameters for assessing nutritional and physiological conditions [71,72]. The dietary inclusion of various levels of JMM did not alter the biological indices of red sea bream in the present study. Likewise, Khosravi et al. [38] explained that supplemented feed attractants (krill hydrolysates, shrimp, and tilapia) in low-FM diets did not alter the K, VSI, and HSI of red sea bream. Unlike in this study, however, the inclusion of mussel meal as a feed stimulant in the feed of common sole (*Solea solea*) altered the K, VSI, and HSI [73]. 

The chemical composition of fish has been reported to be affected by a variety of endogenous and exogenous parameters, such as fish species, size (age), water temperature, nutrient composition, and feed consumption [74,75]. However, the dietary inclusion of JMM did not alter the chemical composition of the whole body of red sea bream in this study, which was consistent with the findings of previous studies [62,76] reporting that supplementation with krill and tuna hydrolysates and squid liver paste in the feeds of red sea bream and GIFT tilapia, respectively, did not affect the chemical composition of the whole fish body. Likewise, the dietary inclusion of multiple stimulants did not change the proximate composition of carp [60]. 

When developing diets with feed stimulants to improve the feed intake and growth (productivity) of fish, it is crucial to evaluate the feasibility of incorporating feed stimulants in practical feeding. Even though the price of the experimental diets tended to increase with the increase in the JMM inclusion levels in this study, and the highest ECR (i.e., the feed cost to produce 1 kg of fish) was obtained for the JMM_100_ diet, a remarkably greater EPI could be achieved for the JMM_40_, JMM_60_, and JMM_100_ diets compared to that for all other diets. The EPI is a proper parameter for determining economic profitability, as it considers growth performance, feed consumption, feed cost, and the selling price of fish [52,77]. Therefore, the results of the economic analysis in this study indicated that including more than 40% JMM in a 60%-FM-based diet for red sea bream is expected to yield greater economic returns for farmers.

## 5. Conclusions

Among 18 crude protein sources, JMM showed the strongest attractiveness to red sea bream. The JMM_40_, JMM_60_, and JMM_100_ diets achieved significantly greater weight gain, SGR, and feed consumption in red sea bream, and they had a greater EPI than that of all other diets. Therefore, the dietary inclusion of JMM at a proportion of more than 40% of FM is highly recommended for practical feed formulation for red sea bream. The incorporation of JMM in the development of low-FM diets for red sea bream will be very helpful for sustainable fish culture in the future.

## Figures and Tables

**Figure 1 animals-14-00771-f001:**
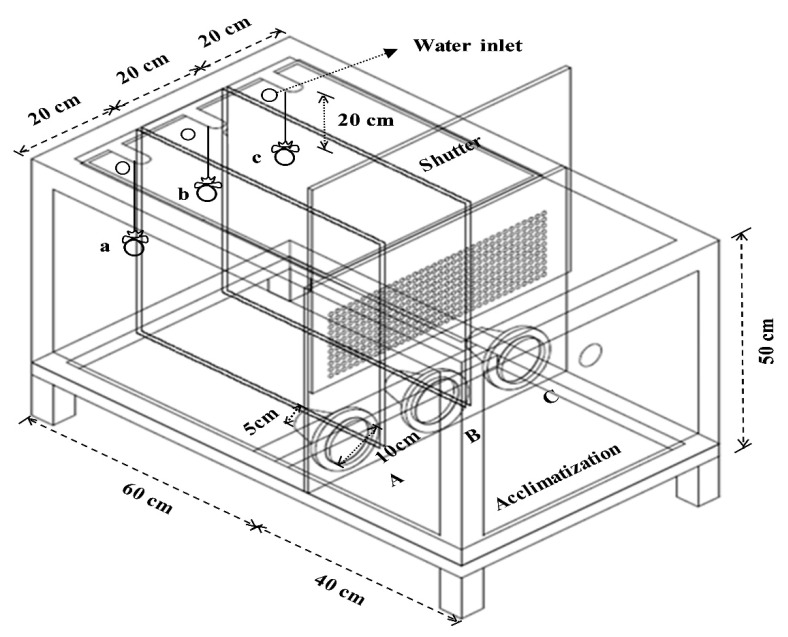
Drawing of a tank used to evaluate red sea bream’s attraction to protein sources and the experimental diets (a, b, and c are the locations of different diets in each attracting chamber; A, B, and C indicate the entrances of each attracting chamber).

**Figure 2 animals-14-00771-f002:**
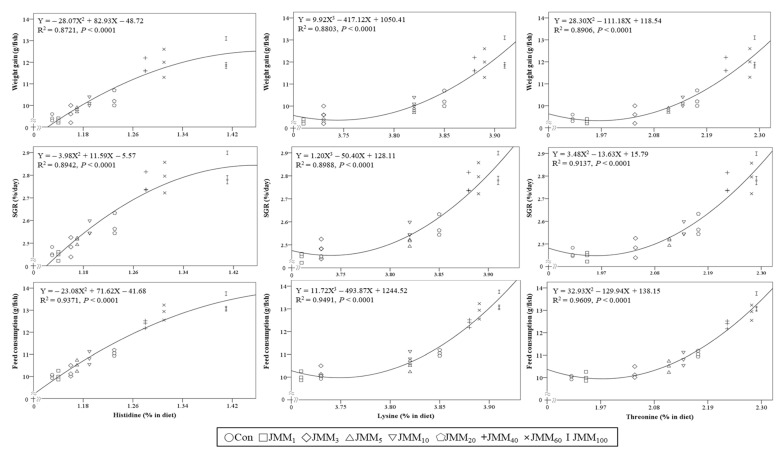
Relationships between the contents of some essential amino acids (histidine, lysine, and threonine) (X) in the experimental diets and the weight gain, SGR, and feed consumption of red sea bream (Y) at the end of the 8-week feeding trial.

**Figure 3 animals-14-00771-f003:**
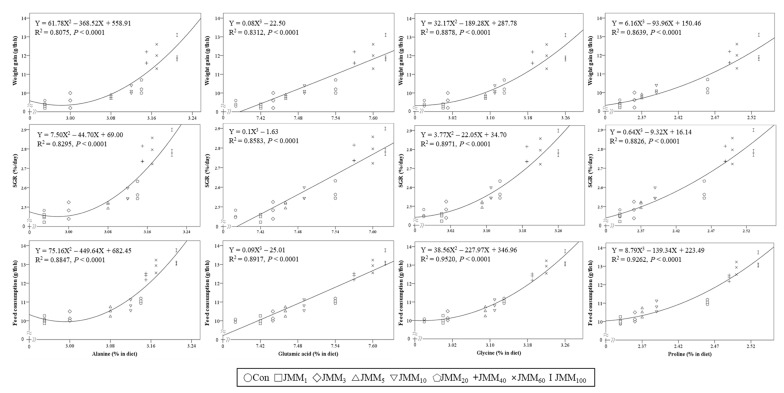
Relationships between the contents of non-essential amino acids (alanine, glutamic acid, glycine, and proline) (X) in the experimental diets and the weight gain, SGR, and feed consumption of red sea bream (Y) at the end of the 8-week feeding trial.

**Table 1 animals-14-00771-t001:** Chemical composition (%, DM basis) and suppliers of protein sources.

	Protein Sources	Chemical Composition				Supply (Nation)
		Moisture	Crude Protein	Crude Lipid	Ash	
Fish meal	Anchovy meal	8.2	73.9	8.1	14.6	Blumar (Punta Arenas, Chile)
	Hydrolyzed fish meal	1.7	74.6	8.6	5.3	Sopropêche (Wimille, France)
	Jack mackerel meal	7.1	73.8	9.1	13.3	Alimentos marinos S.A (Providencia, Chile)
	Pollock meal	6.8	71.2	7.9	17.0	Kodiak Fish Meal Company (Kodiak, AK, USA)
	Sardine meal	7.3	72.6	9.2	15.4	Blumar (Chile)
	Tuna byproduct meal	3.1	63.1	8.5	20.3	Woojin Feed (Gyeonggi, Republic of Korea)
Crustacean meal	Crab meal	1.2	44.1	1.4	40.3	Garunara (Seoul, Republic of Korea)
	Krill meal	8.4	68.5	4.9	11.7	Aker Biomarine (Lysaker, Norway)
	Shrimp meal	3.7	55.8	7.8	20.7	Fortidex S.A. (Guayaquil, Ecuador)
Mollusk meal	Mussel meal	6.7	61.9	9.3	6.4	Garunara (Republic of Korea)
	Squid meal	7.0	70.7	1.9	5.1	APM Logis (Gyeonggi-do, Republic of Korea)
	Squid liver meal	1.9	47.5	19.7	7.2	Dongwoo Ind. (Seoul, Republic of Korea)
Animal by-product meal	Chicken byproduct meal	1.8	63.2	9.4	24.8	Woosin Food (Seoul, Republic of Korea)
	Meat meal	3.2	68.2	12.8	12.6	Hwasong Ind. (Gyeonggi, Republic of Korea)
Plant protein meal	Corn gluten meal	4.5	69.4	1.0	2.5	Hyunjin Ind. (Seoul, Republic of Korea)
	Corn protein concentrate	4.3	79.3	2.8	1.0	Cargill (Minneapolis, MN, USA)
	Fermented soybean meal	3.3	57.9	0.7	7.4	CJ Feed (Seoul, Republic of Korea)
	Soy protein concentrate	3.9	65.6	0.4	6.3	Solae LLC. (St. Louis, MO, USA)

**Table 2 animals-14-00771-t002:** Ingredients and chemical compositions of the experimental diets (%, DM basis).

	Experimental Diets								
	Con	JMM_1_	JMM_3_	JMM_5_	JMM_10_	JMM_20_	JMM_40_	JMM_60_	JMM_100_
Ingredients (%)									
Fish meal (FM) ^a^	60.0	59.4	58.2	57.0	54.0	48.0	36.0	24.0	0
Jack mackerel meal (JMM) ^b^	0	0.6	1.8	3.0	6.0	12.0	24.0	36.0	60.0
Fermented soybean meal	17.5	17.5	17.5	17.5	17.0	16.5	15.5	14.5	12.5
Wheat flour	14.4	14.4	14.3	14.3	14.6	14.8	15.2	15.6	16.4
Fish oil	5.6	5.6	5.7	5.8	5.9	6.2	6.8	7.4	8.6
Vitamin premix ^c^	1.0	1.0	1.0	1.0	1.0	1.0	1.0	1.0	1.0
Mineral premix ^d^	1.0	1.0	1.0	1.0	1.0	1.0	1.0	1.0	1.0
Choline	0.5	0.5	0.5	0.5	0.5	0.5	0.5	0.5	0.5
Nutrients (%)									
Dry matter	98.3	98.1	98.6	98.1	98.0	97.7	98.5	98.2	98.6
Crude protein	52.1	52.5	52.6	52.7	52.5	52.1	52.0	52.6	52.9
Crude lipid	15.1	15.2	15.0	15.0	15.1	15.0	15.1	15.1	15.2
Ash	10.9	10.6	10.8	10.7	10.7	10.7	10.8	10.8	10.7

^a^ Fish meal (FM) (crude protein: 66.9%, crude lipid: 14.5%, ash: 15.8%) imported from Peru was purchased from Daekyung Oil and Transportation Co., Ltd. (Busan, Korea) [USD 2.26/kg FM, USD1 = 1232 Won (Korean currency)]. ^b^ Jack mackerel meal (JMM) (crude protein: 72.8%, crude lipid: 9.8%, ash: 15.4%) imported from Chile was purchased from Daekyung Oil and Transportation Co., Ltd. (Busan, Korea) (USD 2.43/kg JMM). ^c^ Vitamin premix contained the following amounts, which were diluted in cellulose (g/kg mix): L-ascorbic acid, 121.2; DL-α-tocopheryl acetate, 18.8; thiamine hydrochloride, 2.7; riboflavin, 9.1; pyridoxine hydrochloride, 1.8; niacin, 36.4; Ca-D-pantothenate, 12.7; myo-inositol, 181.8; D-biotin, 0.27; folic acid, 0.68; p-aminobenzoic acid, 18.2; menadione, 1.8; retinyl acetate, 0.73; cholecalciferol, 0.003; cyanocobalamin, 0.003. ^d^ Mineral premix contained the following ingredients (g/kg mix): MgSO_4_. 7H_2_O, 80.0; NaH_2_PO_4_. 2H_2_O, 370.0; KCl, 130.0; ferric citrate, 40.0; ZnSO_4_. 7H_2_O, 20.0; Ca-lactate, 356.5; CuCl, 0.2; AlCl_3_. 6H_2_O, 0.15; KI, 0.15; Na_2_Se_2_O_3_, 0.01; MnSO_4_. H_2_O, 2.0; CoCl_2_. 6H_2_O, 1.0.

**Table 3 animals-14-00771-t003:** Attractiveness (%) of various protein sources to red sea bream (*Pagrus major*) with the elapsed time.

		Elapsed Time (minute)		
Test	Protein Sources	10	20	30
1st	Jack mackerel meal	45.3 ± 2.67 ^a^	49.3 ± 1.33 ^a^	52.0 ± 4.00 ^a^
	Krill meal	20.0 ± 2.31 ^bc^	28.0 ± 4.62 ^b^	30.7 ± 5.33 ^ab^
	Soy protein concentrate	8.0 ± 2.31 ^c^	13.3 ± 4.81 ^b^	14.7 ± 5.81 ^bc^
	FSA ^a^	26.7 ± 2.67 ^b^	9.3 ± 4.81 ^b^	2.7 ± 2.67 ^c^
	*p*-value	*p* < 0.0001	*p* < 0.0001	*p* < 0.0001
2nd	Anchovy meal	37.3 ± 1.33 ^a^	41.3 ± 1.33 ^a^	42.7 ± 2.67 ^a^
	Shrimp meal	22.7 ± 1.33 ^b^	29.3 ± 2.67 ^b^	30.7 ± 3.53 ^ab^
	Chicken byproduct meal	17.3 ± 1.33 ^b^	18.7 ± 2.67 ^c^	22.7 ± 3.53 ^b^
	FSA ^a^	22.7 ± 1.33 ^b^	10.7 ± 1.33 ^c^	4.0 ± 2.31 ^c^
	*p*-value	*p* < 0.0001	*p* < 0.0001	*p* < 0.0001
3rd	Sardine meal	37.3 ± 1.33 ^a^	45.3 ± 3.53 ^a^	46.7 ± 4.81 ^a^
	Crab meal	21.3 ± 3.53 ^b^	24.0 ± 2.31 ^b^	25.3 ± 2.67 ^b^
	Meat meal	18.7 ± 1.33 ^b^	28.0 ± 2.31 ^b^	28.0 ± 2.31 ^b^
	FSA ^a^	22.7 ± 3.53 ^b^	2.7 ± 1.33 ^c^	0.0 ± 0.00 ^c^
	*p*-value	*p* < 0.0001	*p* < 0.0001	*p* < 0.0001
4th	Pollock meal	25.3 ± 2.67 ^b^	29.3 ± 2.67 ^b^	33.3 ± 1.33 ^b^
	Squid liver meal	40.0 ± 2.31 ^a^	44.0 ± 0.00 ^a^	44.0 ± 0.00 ^a^
	Fermented soybean meal	18.7 ± 1.33 ^b^	22.7 ± 1.33 ^b^	22.7 ± 1.33 ^c^
	FSA ^a^	16.0 ± 4.62 ^b^	4.0 ± 2.31 ^c^	0.0 ± 0.00 ^d^
	*p*-value	*p* < 0.0001	*p* < 0.0001	*p* < 0.0001
5th	Tuna byproduct meal	37.3 ± 3.53 ^a^	42.7 ± 2.67 ^a^	45.3 ± 1.33 ^a^
	Mussel meal	22.7 ± 2.67 ^ab^	26.7 ± 1.33 ^b^	26.7 ± 1.33 ^b^
	Corn gluten meal	22.7 ± 2.67 ^ab^	24.0 ± 2.31 ^b^	26.7 ± 1.33 ^b^
	FSA ^a^	17.3 ± 4.81 ^b^	6.7 ± 3.53 ^c^	1.3 ± 1.33 ^c^
	*p*-value	*p* < 0.0001	*p* < 0.0001	*p* < 0.0001
6th	Hydrolyzed fish meal	40.0 ± 0.00 ^a^	45.3 ± 2.67 ^a^	48.0 ± 4.62 ^a^
	Squid meal	24.0 ± 2.31 ^b^	28.0 ± 2.31 ^b^	29.3 ± 1.33 ^b^
	Corn protein concentrate	17.3 ± 1.33 ^b^	17.3 ± 1.33 ^bc^	18.7 ± 1.33 ^bc^
	FSA ^a^	18.7 ± 2.67 ^b^	9.3± 2.67 ^c^	4.0 ± 2.31 ^c^
	*p*-value	*p* < 0.0001	*p* < 0.0001	*p* < 0.0001
7th	Jack mackerel meal	37.3 ± 8.11 ^a^	46.7 ± 6.67 ^a^	46.7 ± 6.67 ^a^
	Squid liver meal	21.3 ± 1.33 ^b^	26.7 ± 1.33 ^ab^	28.0 ± 2.31 ^ab^
	Anchovy meal	17.3 ± 2.67 ^b^	20.0 ± 4.00 ^b^	20.0 ± 4.00 ^bc^
	FSA ^a^	24.0 ± 4.62 ^b^	6.7 ± 1.33 ^b^	5.3 ± 1.33 ^c^
	*p*-value	*p* < 0.0001	*p* < 0.0001	*p* < 0.0001
8th	Sardine meal	38.7 ± 5.81 ^a^	46.7 ± 4.81 ^a^	49.3 ± 5.33 ^a^
	Tuna byproduct meal	21.3 ± 1.33 ^ab^	25.3 ± 2.67 ^b^	26.7 ± 3.53 ^b^
	Hydrolyzed fish meal	10.7 ± 3.53 ^b^	13.3 ± 4.81 ^b^	13.3 ± 4.81 ^b^
	FSA ^a^	29.3 ± 3.53 ^ab^	14.7± 1.33 ^b^	10.7 ± 3.53 ^b^
	*p*-value	*p* < 0.0001	*p* < 0.0001	*p* < 0.0001
9th	Jack mackerel meal	37.3 ± 1.33 ^a^	41.3 ± 3.53 ^a^	41.3 ± 3.53 ^a^
	Squid liver meal	18.7 ± 1.33 ^b^	22.7 ± 1.33 ^bc^	26.7 ± 3.53 ^b^
	Sardine meal	24.0 ± 2.31 ^b^	25.3 ± 1.33 ^b^	28.0 ± 2.31 ^b^
	FSA ^a^	20.0 ± 4.00 ^b^	10.7 ± 3.53 ^c^	4.0 ± 2.31 ^c^
	*p*-value	*p* < 0.0001	*p* < 0.0001	*p* < 0.0001
10th	Jack mackerel meal	46.7 ± 1.33 ^a^	50.7 ± 1.33 ^a^	53.3 ± 2.67 ^a^
	Tuna byproduct meal	10.7 ± 1.33 ^c^	12.0 ± 2.31 ^c^	13.3 ± 2.67 ^c^
	Sardine meal	24.0 ± 2.31 ^b^	26.7 ± 1.33 ^b^	29.3 ± 3.53 ^b^
	FSA ^a^	18.7 ± 3.53 ^b^	10.7 ± 1.33 ^c^	4.0 ± 2.31 ^d^
	*p*-value	*p* < 0.0001	*p* < 0.0001	*p* < 0.0001

Values (means of triplicate ± SE) in the same column sharing the same superscript letter are not significantly different (*p* > 0.05). ^a^ FSA is the percentage of fish that stayed in the acclimatization chamber after the 30-min exposure to each protein source.

**Table 4 animals-14-00771-t004:** Attractiveness (%) of the experimental diets to red sea bream (*Pagrus major*) with the elapsed time.

		Elapsed Time (minute)		
Test	Experimental Diets	10	20	30
1st	Con	13.3 ± 1.92 ^c^	15.6 ± 1.11 ^c^	16.7 ± 0.00 ^c^
	JMM_5_	18.9 ± 1.11 ^c^	22.2 ± 1.11 ^bc^	26.7 ± 0.00 ^b^
	JMM_40_	28.9 ± 1.11 ^b^	36.7 ± 1.92 ^a^	38.9 ± 2.22 ^a^
	FSA ^a^	38.9 ± 2.22 ^a^	25.6 ± 1.11 ^b^	17.8 ± 2.22 ^c^
	*p*-value	*p* < 0.0001	*p* < 0.0001	*p* < 0.0001
2nd	JMM_1_	8.9 ± 2.94 ^c^	11.1 ± 2.94 ^c^	12.2 ± 4.01 ^c^
	JMM_10_	22.2 ± 1.11 ^b^	26.7 ± 1.92 ^b^	27.8 ± 2.22 ^b^
	JMM_60_	43.3 ± 1.92 ^a^	51.1 ± 1.11 ^a^	56.7 ± 0.00 ^a^
	FSA ^a^	25.6 ± 4.01 ^b^	11.1 ± 2.94 ^c^	3.3 ± 1.92 ^d^
	*p*-value	*p* < 0.0001	*p* < 0.0001	*p* < 0.0001
3rd	JMM_3_	10.0 ± 3.33 ^b^	15.6 ± 4.44 ^b^	18.9 ± 4.44 ^b^
	JMM_20_	22.2 ± 1.11 ^b^	25.6 ± 1.11 ^b^	26.7 ± 0.00 ^b^
	JMM_100_	43.3 ± 1.92 ^a^	47.8 ± 1.11 ^a^	51.1 ± 1.11 ^a^
	FSA ^a^	24.4 ± 4.84 ^b^	11.1 ± 2.94 ^b^	3.3 ± 3.33 ^c^
	*p*-value	*p* < 0.0001	*p* < 0.0001	*p* < 0.0001
4th	JMM_40_	12.2 ± 2.22 ^b^	14.4 ± 1.11 ^b^	16.7 ± 3.33 ^b^
	JMM_60_	27.8 ± 2.94 ^a^	30.0 ± 3.33 ^a^	35.6 ± 4.01 ^a^
	JMM_100_	34.4 ± 1.11 ^a^	38.9 ± 1.11 ^a^	40.0 ± 1.92 ^a^
	FSA ^a^	25.6 ± 1.11 ^a^	16.7 ± 3.33 ^b^	7.8 ± 4.01 ^b^
	*p*-value	*p* < 0.0001	*p* < 0.0001	*p* < 0.0001

Values (means of triplicate ± SE) in the same column sharing the same superscript letter are not significantly different (*p* > 0.05). ^a^ FSA is the percentage of fish that stayed in the acclimatization chamber after the 30-min exposure to each diet.

**Table 5 animals-14-00771-t005:** Amino acid profiles (% of the diet) of the experimental diets.

				Experimental Diets								
	FM	JMM	Requirement	Con	JMM_1_	JMM_3_	JMM_5_	JMM_10_	JMM_20_	JMM_40_	JMM_60_	JMM_100_
Essential amino acids (%)												
Arginine	4.08	3.96	2.37 ^c^	3.10	3.05	3.03	3.10	3.08	3.10	3.04	3.04	3.03
Histidine	1.50	2.26		1.13	1.14	1.16	1.17	1.19	1.23	1.28	1.31	1.41
Isoleucine	2.70	2.75		2.16	2.14	2.17	2.12	2.14	2.08	2.10	2.10	2.15
Leucine	4.87	4.86		3.85	3.86	3.87	3.88	3.87	3.87	3.81	3.81	3.84
Lysine	5.14	5.28	1.79 ^d^	3.73	3.71	3.73	3.82	3.82	3.85	3.88	3.89	3.91
Methionine	1.85	1.83		1.32	1.29	1.27	1.31	1.30	1.33	1.30	1.31	1.32
Phenylalanine	2.73	2.60		2.17	2.17	2.16	2.15	2.14	2.07	2.05	2.02	2.00
Threonine	2.67	2.85		1.91	1.94	2.04	2.11	2.14	2.17	2.23	2.28	2.29
Tryptophan	0.47	0.47		0.41	0.43	0.43	0.40	0.39	0.40	0.40	0.38	0.38
Valine	3.62	3.40	0.90 ^e^	2.87	2.84	2.84	2.80	2.78	2.73	2.70	2.66	2.61
∑EAA ^a^	29.63	30.26		22.65	22.57	22.70	22.86	22.85	22.83	22.79	22.80	22.94
Non-essential amino acids (%)												
Alanine	4.06	4.49		2.95	2.95	3.00	3.08	3.12	3.14	3.15	3.17	3.21
Aspartic acid	6.05	5.99		4.80	4.75	4.71	4.75	4.74	4.69	4.71	4.72	4.71
Cysteine	0.70	0.80		0.72	0.73	0.76	0.76	0.76	0.76	0.77	0.77	0.79
Glutamic acid	8.48	8.73		7.38	7.42	7.44	7.46	7.49	7.54	7.57	7.60	7.62
Glycine	4.11	4.68		2.96	3.00	3.01	3.09	3.11	3.13	3.19	3.22	3.26
Proline	2.79	3.16		2.34	2.34	2.36	2.37	2.39	2.46	2.49	2.50	2.53
Serine	2.27	2.32		1.62	1.60	1.64	1.65	1.65	1.66	1.67	1.73	2.02
Tyrosine	1.13	1.21		0.88	0.91	0.91	0.92	0.92	0.94	1.03	1.06	1.34
∑NEAA ^b^	29.59	31.38		23.65	23.70	23.83	24.08	24.18	24.32	24.58	24.77	25.48

^a^ ∑EAA: total content of essential amino acids. ^b^ ∑NEAA: total content of non-essential amino acids. Arginine ^c^, lysine ^d^, and valine ^e^ requirements were obtained from [53,54,55], respectively.

**Table 6 animals-14-00771-t006:** Fatty acid profiles (% of total fatty acids) of the experimental diets containing the various concentrations of jack mackerel meal (JMM).

				Experimental Diets								
	FM	JMM	Requirement	Con	JMM_1_	JMM_3_	JMM_5_	JMM_10_	JMM_20_	JMM_40_	JMM_60_	JMM_100_
C10:0	0.11	0.06		0.06	0.06	0.05	0.06	0.05	0.03	0.04	0.03	0.03
C12:0	0.07	0.05		0.04	0.04	0.04	0.04	0.04	0.04	0.04	0.03	0.03
C14:0	6.89	5.01		4.04	3.95	3.85	3.77	3.71	3.59	3.28	2.79	2.19
C16:0	20.22	21.86		16.70	16.74	16.80	16.98	17.09	17.08	17.20	17.31	17.40
C18:0	5.67	7.30		3.35	3.42	3.58	3.63	3.72	3.85	4.22	4.72	5.61
C20:0	0.12	0.17		0.17	0.20	0.22	0.21	0.18	0.22	0.17	0.20	0.22
C24:0	1.65	2.31		0.37	0.36	0.44	0.49	0.50	0.51	0.62	0.77	1.08
∑SFA ^a^	34.73	36.76		24.73	24.77	24.98	25.18	25.29	25.32	25.57	25.85	26.56
C14:1n-5	0.20	0.14		0.09	0.09	0.09	0.09	0.09	0.08	0.07	0.06	0.04
C15:1n-7	0.08	0.07		0.04	0.04	0.04	0.04	0.04	0.04	0.04	0.03	0.03
C16:1n-7	9.01	6.74		5.54	5.40	5.34	5.25	5.16	5.02	4.79	4.11	3.40
C17:1n-1	0.49	0.86		0.16	0.16	0.16	0.16	0.16	0.16	0.17	0.17	0.17
C18:1n-9	18.64	20.87		25.17	25.22	25.25	25.41	25.61	25.82	25.95	26.93	27.72
C20:1n-9	3.01	1.80		3.97	3.87	3.71	3.68	3.49	3.44	2.74	2.24	1.17
C22:1n-9	1.72	0.83		3.27	3.16	3.00	2.95	2.87	2.76	2.52	2.23	1.55
C24:1n-9	1.03	0.38		0.07	0.07	0.07	0.07	0.07	0.05	0.05	0.04	0.04
∑MUFA ^b^	34.18	31.69		38.31	38.01	37.66	37.65	37.49	37.37	36.33	35.81	34.12
C18:2n-6	0.96	1.73		20.84	21.07	21.15	21.29	21.88	22.05	22.39	22.56	22.96
C18:3n-3	0.52	0.76		2.88	2.93	2.94	2.83	2.88	3.02	3.18	3.37	3.62
C18:3n-6	0.07	0.10		0.08	0.08	0.08	0.08	0.08	0.09	0.09	0.09	0.10
C20:2n-6	0.15	0.17		0.14	0.14	0.14	0.14	0.14	0.14	0.13	0.14	0.14
C20:4n-6	0.44	0.79		0.26	0.27	0.28	0.28	0.28	0.29	0.33	0.34	0.40
C20:5n-3	13.73	10.96	6.62 ^d^	5.09	4.84	4.82	4.77	4.67	4.54	4.37	4.12	4.02
C22:6n-3	11.91	13.89	3.31 ^d^	4.69	4.88	5.50	5.88	6.08	6.18	6.24	6.29	6.44
DHA:EPA ratio	0.87	1.27		0.92	1.01	1.14	1.23	1.30	1.36	1.43	1.53	1.60
∑n-3 HUFA ^c^	25.64	24.85		9.78	9.72	10.32	10.65	10.75	10.72	10.61	10.41	10.46
Unknown	3.31	3.15		2.98	3.01	2.45	1.90	1.21	1.00	1.37	1.43	1.64

^a^ ∑SFA: total content of saturated fatty acids. ^b^ ∑MUFA: total content of monounsaturated fatty acids. ^c^ ∑ n-3 HUFA: total content of n-3 highly unsaturated fatty acids. ^d^ EPA and DHA requirements were obtained from [40].

**Table 7 animals-14-00771-t007:** Initial weight (g/fish), final weight (g/fish), survival, weight gain (g/fish), and specific growth rate (SGR, %/day) of red sea bream (*Pagrus major*) fed the experimental diets containing various levels of jack mackerel meal (JMM) for 8 weeks.

	Experimental Diets									SEM	ANOVA *p*-Value	Regression Analysis		
	Con	JMM_1_	JMM_3_	JMM_5_	JMM_10_	JMM_20_	JMM_40_	JMM_60_	JMM_100_			Model	R^2^	*p*-Value
Initial weight (g/fish)	3.2	3.2	3.2	3.2	3.2	3.2	3.2	3.2	3.2	0.01				
Final weight (g/fish)	12.6 ^b^	12.5 ^b^	12.8 ^b^	13.0 ^b^	13.3 ^b^	13.5 ^b^	15.0 ^a^	15.1 ^a^	15.4 ^a^	0.23	*p* < 0.0001	Q	0.9049	*p* < 0.0001
Survival (%)	97.8	97.8	96.7	95.6	94.4	94.4	95.6	93.3	96.7	0.68	*p* > 0.8	NR		
Weight gain (g/fish)	9.4 ^b^	9.3 ^b^	9.6 ^b^	9.8 ^b^	10.1 ^b^	10.3 ^b^	11.8 ^a^	12.0 ^a^	12.2 ^a^	0.27	*p* < 0.0001	Q	0.8992	*p* < 0.0001
SGR (%/day) ^a^	2.46 ^bc^	2.44 ^c^	2.48 ^bc^	2.51 ^bc^	2.56 ^bc^	2.58 ^b^	2.76 ^a^	2.79 ^a^	2.82 ^a^	0.029	*p* < 0.0001	Q	0.9210	*p* < 0.0001

Values (means of triplicate experiments) in the same row sharing a superscript letter are not significantly different (*p* > 0.05). Abbreviation: SEM, pooled standard error of means; Q, quadratic; NR, no relationship. ^a^ SGR (%/day) = (Ln final weight of fish − Ln initial weight of fish) × 100/days of feeding trial.

**Table 8 animals-14-00771-t008:** Feed consumption, feed efficiency (FE), protein efficiency ratio (PER), protein retention (PR), condition factor (K), viscerosomatic index (VSI), and hepatosomatic index (HSI) of red sea bream (*Pagrus major*) fed the experimental diets containing various levels of jack mackerel meal (JMM) for 8 weeks.

	Experimental Diets									SEM	ANOVA *p*-Value	Regression Analysis		
	Con	JMM_1_	JMM_3_	JMM_5_	JMM_10_	JMM_20_	JMM_40_	JMM_60_	JMM_100_			Model	R^2^	*p*-Value
Feed consumption (g/fish)	10.0 ^e^	10.0 ^e^	10.2 ^de^	10.5 ^cde^	10.8 ^cd^	11.1 ^c^	12.4 ^b^	12.9 ^ab^	13.3 ^a^	0.24	*p* < 0.0001	Q	0.9635	*p* < 0.0001
FE ^a^	0.94	0.93	0.94	0.94	0.94	0.94	0.96	0.93	0.92	0.005	*p* > 0.6	NR		
PER ^b^	1.81	1.77	1.79	1.77	1.79	1.79	1.83	1.75	1.74	0.009	*p* > 0.4	NR		
PR (%) ^c^	28.5	28.4	27.8	27.9	27.6	29.8	30.4	29.3	28.6	0.38	*p* > 0.7	NR		
K (g/cm^3^) ^d^	1.86	1.86	1.85	1.85	1.85	1.86	1.88	1.86	1.88	0.011	*p* > 0.8	NR		
VSI (%) ^e^	6.83	6.91	7.02	6.89	6.93	6.89	6.82	6.96	6.94	0.063	*p* > 0.9	NR		
HSI (%) ^f^	1.39	1.42	1.39	1.40	1.47	1.42	1.41	1.39	1.39	0.019	*p* > 0.9	NR		

Values (means of triplicate experiments) in the same row sharing a superscript letter are not significantly different (*p* > 0.05). Abbreviation: SEM, pooled standard error of means; Q, quadratic; NR, no relationship. ^a^ Feed efficiency (FE) = [Total final weight (g) − total initial weight (g) + total weight of dead fish (g)]/total feed consumption (g). ^b^ Protein efficiency ratio (PER) = Weight gain of fish/protein consumption of fish. ^c^ Protein retention (PR, %) = Protein gain of fish × 100/protein consumption of fish. ^d^ Condition factor (K, g/cm^3^) = Body weight of fish (g) × 100/total length of fish (cm)^3^. ^e^ Visceorosomatic index (VSI, %) = Viscera weight of fish (g) × 100/body weight of fish (g). ^f^ Hepatosomatic index (HSI, %) = Liver weight of fish (g) × 100/body weight of fish (g).

**Table 9 animals-14-00771-t009:** Chemical composition (% of wet weight) of the whole body of red sea bream (*Pagrus major*) fed the experimental diets containing various levels of jack mackerel meal (JMM) for 8 weeks.

	Experimental Diets									SEM	ANOVA *p*-Value	Regression Analysis		
	Con	JMM_1_	JMM_3_	JMM_5_	JMM_10_	JMM_20_	JMM_40_	JMM_60_	JMM_100_			Model	R^2^	*p*-Value
Moisture	74.9	75.3	75.4	76.2	75.2	72.9	73.3	74.2	75.0	0.33	*p* > 0.3	NR		
Crude protein	15.5	15.7	15.3	15.5	15.3	15.7	15.9	16.0	15.6	0.10	*p* > 0.8	NR		
Crude lipid	9.4	9.6	9.2	9.3	9.3	9.3	9.6	9.6	9.6	0.09	*p* > 0.9	NR		
Ash	4.7	4.9	5.0	5.2	5.0	4.5	4.7	4.8	4.5	0.09	*p* > 0.7	NR		

Values (means of triplicate) in the same row are not significantly different (*p* > 0.05). Abbreviation: SEM, pooled standard error of means; NR, no relationship.

**Table 10 animals-14-00771-t010:** Diet price (USD/kg), economic conversion ratio (ECR, USD/kg), and economic profit index (EPI, USD/fish) of the experimental diets.

	Experimental Diets									SEM	ANOVA *p*-Value	Regression Analysis		
	Con	JMM_1_	JMM_3_	JMM_5_	JMM_10_	JMM_20_	JMM_40_	JMM_60_	JMM_100_			Model	R^2^	*p*-Value
Diet price (USD/kg)	1.87	1.87	1.87	1.88	1.88	1.90	1.93	1.96	20.3					
ECR (USD/kg) ^a^	1.49 ^d^	1.50 ^d^	1.50 ^d^	1.52 ^cd^	1.53 ^cd^	1.56 ^cd^	1.60 ^bc^	1.68 ^ab^	1.75 ^a^	0.018	*p* < 0.003	L	0.8935	*p* < 0.0001
EPI (USD/fish) ^b^	0.24 ^b^	0.23 ^b^	0.24 ^b^	0.24 ^b^	0.25 ^b^	0.25 ^b^	0.28 ^a^	0.28 ^a^	0.29 ^a^	0.004	*p* < 0.0001	Q	0.8792	*p* < 0.0001

Values (means of triplicate experiments) in the same row sharing a superscript letter are not significantly different (*p* > 0.05). Abbreviations: SEM, pooled standard error of means; L, linear; Q, quadratic; NR, no relationship. ^a^ Economic conversion ratio (ECR, USD/kg) = feed consumption of fish (kg/fish)/[weight gain of fish (kg/fish) × diet price (USD/kg)]. ^b^ Economic profit index (EPI, USD/fish) = [final weight of fish (kg/fish) × selling price of fish (20.29 USD/kg)] − [feed consumption of fish (kg/fish) × diet price (USD/kg)].

## Data Availability

The data are available from the corresponding author upon reasonable request.

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
