# Peer review of "Effects of Dietary Inclusion of a Crude Protein Source Exhibiting the Strongest Attractiveness to Red Sea Bream (Pagrus major) on Growth, Feed Availability, and Economic Efficiency"

_animals, 2024, doi:10.3390/ani14050771_

Round 1

Reviewer 1 Report

Comments and Suggestions for Authors

In this research, the author wants to test several sources of raw materials for fish feed. The research carried out is quite interesting. Unfortunately, the research is still unable to explain the purpose of this research. Does this study try to replace fish meal as a raw material source for feed? Or are you looking for the correct type of fish to make the fish meal?

Apart from that, several parts need to be addressed, as follows:

Abstract:

The research objectives must be stated clearly

Method:

Your statement in 2.1.2 is, "Then, the top four selected protein sources (9th – 10th test) were chosen to finally elucidate the strongest attractiveness of the crude protein source to juvenile fish." I didn't find any research results; only JMM is available.

2.2.2. In the control, what species did the fishmeal used come from?

2.2.6 how many samples are used for testing amino acids and fatty acids per treatment

Results:

1. The presentation of Table 5 needs to be improved

2. The values in Table 7, Table 8 and Table 9 should be presented in average and Std form

3. The table needs to be presented neatly

Discussion

In this research, you recommend using 100% JMM to increase the attractiveness, growth, and efficiency of fish farming. You believe that even though it requires more capital to make feed, the growth rate will be faster and higher. Unfortunately, I didn't see you discussing it from a conservation perspective, whether using fishmeal at high levels like this won't cause other problems.

Author Response

Reviewer #1

The manuscript has been addressed or revised based on reviewer 1’ comment. All corrections or revisions were highlighted in red-color.

In this research, the author wants to test several sources of raw materials for fish feed. The research carried out is quite interesting. Unfortunately, the research is still unable to explain the purpose of this research. Does this study try to replace fish meal as a raw material source for feed? Or are you looking for the correct type of fish to make the fish meal?

→The purpose of this study is to improve the growth and feed intake of red sea bream by dietary incorporating feed protein source, which exhibit the strongest attractiveness to red sea bream based on feeding behavior (movement of fish toward the attracting chamber after introduction of feed ingredient in Fig. 1). We evaluated attractiveness of 18 crude protein sources to red sea bream and determined optimum inclusion level of jack mackerel meal (JMM) exhibiting the strongest attractiveness to fish. Then we determined dietary optimum inclusion level of JMM based on growth performance and feed availability of red sea bream. Finally, we evaluated economic analysis of the study to help readers to understand how beneficial to farmer when applying dietary treatment we recommended in this study. This will a very useful technique to develop low fish meal (FM) diets without monitoring electroneurophysiology of fish, which was commonly used to look for feed attractants and/or stimulants in the past. Based on results of this study [optimum jack mackerel meal (JMM) inclusion level of 40% of FM in the diet, being equivalent to 24% JMM in the diet], we have developed low FM diets replacing different levels of fish meal with various animal and/or plant protein sources with jack mackerel meal (JMM) inclusion for sustainable fish culture in another reports. We think we explained all clearly in Introduction section.  

Apart from that, several parts need to be addressed, as follows:

Abstract:

The research objectives must be stated clearly

→It was revised as you pointed out.

Method:

Your statement in 2.1.2 is, "Then, the top four selected protein sources (9th – 10th test) were chosen to finally elucidate the strongest attractiveness of the crude protein source to juvenile fish." I didn't find any research results; only JMM is available.

→From 1st – 8th test, top 4 protein sources (JMM, sardine meal, squid liver meal, and tuna by-product meal) with strong attractiveness to red sea bream were identified. Finally, through 9th – 10th test, JMM exhibited the strongest attractiveness to rockfish (Table 3).

2.2.2. In the control, what species did the fishmeal used come from?

→FM used in the Con diet was anchovy meal, which is commonly used as a FM source in commercial fish feeds.

2.2.6 how many samples are used for testing amino acids and fatty acids per treatment

→ Amino and fatty acid profiles of the experimental diets were analyzed without replication.

Results:

  1. The presentation of Table 5 needs to be improved.

→It was revised as you pointed out. When we submitted this manuscript to Animals online, all Tables were clear. However, they had been changed while editing and transferring the current format of the manuscript. Thank you for your comment.

  1. The values in Table 7, Table 8 and Table 9 should be presented in average and Std form

→ Thank you for your suggestion. However, Tables 7, 8, and 9 had not only the results of results the 8-week feeding trial, but also results of regression analysis. To prevent excessive data in these tables, we included SEM (pooled standard error of means) instead of standard error. We listed some studies that presented the results using the same form as ours.

Lee, C.H., Kim, H.S., Lee, K.W., Han, G.S., Byun, S., Lim, H.J., Lee, D.H., Choi, J., 2021. Effects of dietary lipid level on growth performance, feed utilization, fatty composition and antioxidant parameters of juvenile walleye pollock, Gadus chalcogrammus. Aquac. Rep. 19, 100631. https://doi.org/10.1016/j.aqrep.2021.100631.

Novriadi, R., Salze, G., Abebe, A., Hanson, T., Davis, D.A., 2019. Partial or total replacement of fish meal in the diets of Florida pompano Trachinotus carolinus. Aquac. Res. 50, 1527-1538. https://doi.org/10.1111/are.14029.

Liu, T., Han, T., Wang, J., Liu, T., Bian, P., Wang, Y., Cai, X., 2021. Effects of replacing fish meal with soybean meal on growth performance, feed utilization and physiological status of juvenile redlip mullet Liza haematocheila. Aquac. Rep. 20, 100756. https://doi.org/10.1016/j.aqrep.2021.100756.

  1. The table needs to be presented neatly

→All tables were revised as you pointed out.

Discussion

In this research, you recommend using 100% JMM to increase the attractiveness, growth, and efficiency of fish farming. You believe that even though it requires more capital to make feed, the growth rate will be faster and higher. Unfortunately, I didn't see you discussing it from a conservation perspective, whether using fishmeal at high levels like this won't cause other problems.

→Since JMM is also a type of FM, its use in fish feeds may not provide significant benefits from a conservation perspective. However, we have confirmed that JMM acts as a feed enhancer in the 60% FM-based diets of red sea bream in this study in the 8-week feeding trial. In addition, based on growth performance and EPI of the study, we concluded that 100% JMM inclusion was the most recommendable dietary treatment to induce the maximum growth rate and the greatest EPI in the original manuscript. We partially agreed your suggestion and comment on this issue (using high level of fish meal). We revised our results from this study and concluded that dietary inclusion of JMM more than 40% is a practical feed formulation to induce remarkable improvement in growth and economic return to farmer. As mentioned in Discussion section, we will evaluate the inclusion effect of JMM as a feed enhancer in low FM diets in follow-up studies, which will offer significant benefits to farmer from a conservation perspective, particularly concerning sustainable aquaculture.

I really appreciate for your valuable comments on this manuscript.

From corresponding author

Reviewer 2 Report

Comments and Suggestions for Authors

Author Response

Response to Reviewer #2

The manuscript has been addressed or revised based on reviewer 2’ comment. All corrections or revisions were highlighted in red-color.

1/ This study evaluated the economic efficiency of different protein sources in diets for red sea bream aquaculture in Korea. In the methodology section 2.2.7 on the economic analysis of the feeding experiment, the evaluation was conducted using USD as the currency type. However, exchange rates and market prices for both ingredients and fish can fluctuate significantly over time. For example, the costs of fish meal and jack mackerel meal today would likely differ from their pricing at the time of the study. Similarly, the market price per kilogram of red sea bream would be expected to change across seasons and years. These variabilities introduce uncertainty when attempting to definitively conclude whether one protein source is unequivocally the “most suitable” economically for farmers.

→We partially agree with your comment. However, we believe that economic analysis should be performed in the feed nutrition study and it will be helpful for readers to understand how beneficial dietary treatments will produce for farmers. In particularly, as we manipulated JMM in the diet of red sea bream, which is more costly than FM, it was better for us to determine whether the use of JMM instead of FM was economically feasible. Economic analysis of the study will be very helpful for readers to understand how much economic return will be provided to farmers when applying dietary treatments. Although prices of feed ingredients and fish, and exchange rate of USD varied daily, we would endeavor to provide detailed information on their prices and the calculation of economic parameters as much as possible.

2/ A key component not specifically addressed in the methodology is whether the authors checked that dataset distributions satisfied parametric assumptions before conducting statistical analysis. Verifying normality and homogeneity of variance would be prudent given the multiple response variables (e.g. growth, consumption, indices) and diet groups involved. The statistical analysis section is very brief. More details on the specific statistical tests used and how data was analyzed would strengthen this section.

→Thank you for your suggestion. We added some details in statistical analysis of the study including verification of normality and homogeneity of variance in statistical analysis. All dataset are satisfied for normality and homogeneity.

3/ Section 2.2.3, where the authors evaluate how attractive the experimental feeds are to red sea bream, they state that they used the same "procedures and methods" as in the previous crude protein attractiveness tests. However, it's unclear specifically what methods they are referring to.

→ Thank you for your suggestion. Except for the experimental condition (fish size and water temperature) we described in Section 2.2.3, all experimental procedures and methods were the same in Sections 2.1.3 and 2.1.4. In order to avoid redundancy in explanation, they were omitted. However, we added a brief sentence to avoid confusion as you pointed out.

4/ Please carefully check the format for several paragraph at Discussion section. The discussion section could do a better job linking the results back to the original research objectives and questions. There also seems to be some repetition from the results section that could be condensed.

→ Thank you for your suggestion. We condensed some repeated sentences in Results and Discussion sections. Additionally, we revised some sentences in Discussion section to improve readability.

5/ Is there any limitation from this study?

→There are no significant limitations in our study. We conducted our study meticulously, considering various factors to ensure the reliability and validity of our findings. Our experimental design was carefully planned, and we took measures to minimize any potential biases.

6/ The authors state that inclusion of 100% jack mackerel meal (JMM) at the expense of fish meal leads to the greatest growth, feed consumption, and economic return. However, the data shows no significant difference in growth between the 40% JMM, 60% JMM, and 100% JMM diets. Additionally, the authors do not provide any statistical analysis comparing the economic performance of these diets. Given the lack of statistical significance and comparative economic analysis between the 40%, 60% and 100% JMM diets, the data does not seem to strongly support the authors' conclusive statement about the benefits of 100% inclusion. A deeper discussion and analysis contrasting the performance of the 40%, 60% and 100% JMM diets is needed to justify promoting 100% JMM as optimal. The authors should reconsider or reframe their conclusion to better reflect the uncertainties and limitations of the data.

→Thank you for your comment. We confirmed that there is no significant difference in EPI of the JMM40, JMM60, and JMM100 diets. From a conservation perspective, we’d better not use the JMM100 diet. So we revised that the JMM40 diet was the most recommendable dietary treatment because no significant differences in weight gain, SGR, and feed consumption of red sea bream and EPI were found among the JMM40, JMM60, and JMM100 diets in this study. We have revised unclear sentences related to this issue throughout the entire manuscript accordingly.

7/ There is little discussion of the specific compounds or attractive nutritional components in jack mackerel meal that may be driving the increased feed intake and growth.

→In the discussion section, improved feed intake of red sea bream achieved by the dietary inclusion of JMM was mainly explained through the amino acid of JMM. Contents of histidine, alanine, glycine, glutamic acid, and proline, which were relatively higher in JMM compared to FM, were considered to elicit positive feeding responses in red sea bream (Goh and Tamura, 1980; Shimizu et al., 1990; Tola et al., 2019). The relatively higher DHA/EPA ratio in JMM, compared to FM, was also discussed as partially improving the growth of red sea bream. Additionally, it was also discussed that jack mackerel muscle extract contains IMP and lactic acid, which are well known to elicit strong feeding responses in yellowtail.

Goh, Y., Tamura, T., 1980. Olfactory and gustatory responses to amino acids in two marine teleosts—red sea bream and mullet. Comp. Biochem. Physiol. Part C Comp. Pharmacol. 66, 217–224. https://doi.org/10.1016/0306-4492(80)90130-6.

Shimizu, C., Ibrahim, A., Tokoro, T., Shirakawa, Y., 1990. Feeding stimulation in sea bream, Pagrus major, fed diets supplemented with Antarctic krill meals. Aquaculture 89, 43–53. https://doi.org/10.1016/0044-8486(90)90232-C.

Tola, S., Fukada, H., Masumoto, T., 2019. Effects of natural feeding stimulants and glutamic acid supplementation on the feed intake, growth performance and digestive enzyme activities of red sea bream (Pagrus major) fed fish meal‐free soy protein concentrate (SPC)‐based diet. Aquac. Res. 50, 1912–1920. https://doi.org/10.1111/are.14077.

8/ No histological or immune function evaluations are included.

→Since JMM is a type of FM, dietary inclusion of JMM instead of FM is unlikely to have any negative effect on hematological parameters and non-specific immune responses of fish. In support of this, Choi et al. (2020) reported that dietary inclusion of JMM as the main protein source (60% in diet) did not negatively affect the hematological parameters of fish. Including histological changes of fish in this study is too enormous to cover. 

Choi, J., Lee, K.W., Han, G.S., Byun, S.G., Lim, H.J., Kim, H.S., 2020. Dietary inclusion effect of krill meal and various fish meal sources on growth performance, feed utilization, and plasma chemistry of grower walleye pollock (Gadus chalcogrammus, Pallas 1811). Aquac. Rep. 17, 100331. https://doi.org/10.1016/j.aqrep.2020.100331. 

9/ The long-term sustainability and costs of fisheries used for fishmeal production should be discussed as it relates to the environmental impacts of aquaculture feeds.

→Thank you for your suggestion. Unfortunately we had not monitored other environmental parameters, except for water temperature, DO, salinity, and pH. However, we did not include any ingredients in the feed that could potentially have a negative impact on the marine environment. JMM is one of the widely used FM globally indicating that the inclusion of JMM in fish feeds do not have any adverse effects on marine environments. Additionally, we have conducted numerous feeding trials in this experimental condition (flow-through tank systems), but the experimental diets did not negatively affect water quality and performance of fish.

I really appreciate for your valuable comments on this manuscript.

From corresponding author

Reviewer 3 Report

Comments and Suggestions for Authors

This manuscript provided interesting and valuable information for the feed producers and farmers on the attractiveness and usage of protein sources. The introduction provided enough information for the explanation of the research. The materials and method were described in detail. The results are also well presented, except I found that the graphical and mathematical relations in Figures 2 and 3 of specific growth rate, weight gain and feed consumption to some amino acids content in feed have not much sense because the feeds and not differing in one amino acid presented on the graph but in composition of all amino acids. Hence, the effect on growth and feed consumption is not the result of only one amino acid. I think more realistic would be to relate SGR to percentage of jack mackerel meal in feed.

The results were well explained and compared with the literature data. The conclusion was supported by the results obtained.

There is uneven font in some parts of the discussion chapter.

Author Response

Response to Reviewer #3

The manuscript has been addressed or revised based on reviewer 3’ comment. All corrections or revisions were highlighted in red-color.

This manuscript provided interesting and valuable information for the feed producers and farmers on the attractiveness and usage of protein sources. The introduction provided enough information for the explanation of the research. The materials and method were described in detail. The results are also well presented, except I found that the graphical and mathematical relations in Figures 2 and 3 of specific growth rate, weight gain and feed consumption to some amino acids content in feed have not much sense because the feeds and not differing in one amino acid presented on the graph but in composition of all amino acids. Hence, the effect on growth and feed consumption is not the result of only one amino acid. I think more realistic would be to relate SGR to percentage of jack mackerel meal in feed.

→Thank you for your valuable comment. We did not explain that each amino acid (histidine, alanine, glycine, glutamic acid, and proline) we mentioned had positive impacts on growth and feed consumption of fish. Feed consumption of fish is influenced by not only individual amino acids in diet, but also by their interactions among different amino acids. However, it is noteworthy that most of those amino acids (histidine, alanine, glycine, glutamic acid, and proline) have been previously recognized as the feed enhancers in the several studies. Therefore, we tried to show relationships of growth and feed intake of red sea bream vs amino acids (histidine, alanine, glycine, glutamic acid, and proline) in the experimental diets with increased JMM inclusion levels. Additionally, results of regression analysis of parameters we measured and dietary JMM inclusion levels were also presented in Tables 7–10.

There is uneven font in some parts of the discussion chapter.

→ It was revised as you pointed out.

I really appreciate for your valuable comments on this manuscript.

From corresponding author

Round 2

Reviewer 1 Report

Comments and Suggestions for Authors

The author has made significant change in this revised version and make it acceptable for publication. However, in conclusion, the author should recommend a futher research that considers the conservation and suistanabily aspect in fish feed production.

Author Response

Response to Reviewer 1

We appreciate for your valuable comment and suggestion to improve readability of the manuscript.

We added a following sentence in conclusion as you pointed out: Incorporation of JMM to develop the low FM diets of red sea bream will be a very helpful for sustainable fish culture in future.

From corresponding author